# Morphological, Histological and Morphometrical Aspects of Auditory Ossicles in Pig Fetuses (*Sus scrofa domestica*)

**DOI:** 10.3390/ani15081129

**Published:** 2025-04-14

**Authors:** Cristian Olimpiu Martonos, Pompei Bolfa, Andras-Laszlo Nagy, David Hilchie, William Brady Little, Cristian Constantin Dezdrobitu, Alexandru Ion Gudea

**Affiliations:** 1Department of Biomedical Sciences, Ross University School of Veterinary Medicine, Basseterre 00265, Saint Kitts and Nevis; cmartonos@rossvet.edu.kn (C.O.M.); pompeibolfa@gmail.com (P.B.); anagy@rossvet.edu.kn (A.-L.N.); dhilchie@rossvet.edu.kn (D.H.); brlittle@rossvet.edu.kn (W.B.L.); cdezdrobitu@rossvet.edu.kn (C.C.D.); 2Department of Anatomy, Faculty of Veterinary Medicine, University of Agricultural Sciences and Veterinary Medicine, 400372 Cluj-Napoca, Romania

**Keywords:** *Sus scrofa*, ossicula auditoria, fetus, malleus, incus, stapes, lever ratio, gross anatomy, histology, morphometry

## Abstract

This paper approaches the elements of fine morphological details regarding auditory ossicles in a certain stage of fetal development in swine. This paper describes the gross anatomical features of the ossicular chain (malleus, incus and stapes) along with their complex metrical features. The different functional indices were calculated based on the available metrical data and compared to some known data in the literature. This histological investigation brings a series of insights into the existence of cortical fascicles and mineralization observed in the malleus, incus and stapes, as they display different organizations and degrees of compartmentalization and structuring of the bony and cartilaginous tissue.

## 1. Introduction

The morphological and functional features of the ear (*Organum vestibulocochleare*) are extremely complex in mammalian species [1,2]. Anatomically, this sense organ is located within the temporal bone and is essential for hearing and balance [3,4]. This region is comprised of three segments: external ear (*Auris externa*), middle ear (*Auris media*) and internal ear (*Auris interna*) [3,5].

The middle ear is also called the tympanic cavity and has been described as an air-filled cavity located in the tympanic segment of the temporal bone [6,7]. The tympanic cavity is bordered laterally by the tympanic membrane (*Membrana tympany*) and medially by the oval window (*Fenestra vestibuli*). Between these elements, three small ossicles can be identified in mammals: the malleus, incus and stapes. Most lateral is the malleus, which is in direct contact with the internal surface of the tympanic membrane, and most medial is the stapes, which cover the oval window. Lastly, the incus articulates laterally with the malleus and medially with the stapes [8,9].

Most anatomical studies to date have focused on the description of postnatal specimens, while the study of the fetal period is lacking in the scientific literature [2]. Although the study of auditory ossicles is far from complete, there are at least a few published investigations in human fetuses [10,11,12,13,14] whereas the number of studies describing the morphological aspects of fetal auditory ossicles in the veterinary field is severely lacking [1,2].

The embryological development of fetal ossicles in the middle ear of humans is described by two theories: a classical theory of the embryological development of auditory ossicles and a newer alternative theory [12]. The first one describes the embryological origin of the malleus and incus from the first pharyngeal arch or the level of Meckel’s cartilage. According to this theory, the stapes has its origin in the second pharyngeal arch, also named Reichert’s cartilage [13,15,16,17]. The alternative, and likely more precise, theory describes the head of the malleus and the body of the incus originating from the first pharyngeal arch, while the other segments of the malleus, incus and the entire stapes develop from the second pharyngeal arch [13]. All mammals possess these three ossicles, but there are species where the malleus and incus are fused and form a physiological incudomalleolar complex [13,18,19,20,21,22,23,24]. Reptiles and birds have only one acoustic bone (stapes for reptiles and the columella auris for birds) [25].

The pig ear complex is similar to that of people and therefore often utilized and appreciated as an alternative for anatomical dissection of the human temporal bone, and specifically by ear surgeons as a viable model for training and the study of this region [26,27,28]. For this reason, the present study aims to describe the anatomical, histological and morphometrical details of the auditory ossicles in pig fetuses and the comparison of these aspects with the reported data for adult specimens.

## 2. Materials and Methods

### 2.1. Sample Collection and Image Processing Technique

The biological material for this study was provided by the Anatomy Department of Ross University, School of Veterinary Medicine, Saint Kitts and Nevis, during the academic year 2023–2024. Auditory tissue was collected and measured from 8 fetuses of different ages and weights originating from the local slaughterhouse (Saint Kitts Abbatoire, Federal Meat Inspection Act/USDA regulations compliant) resulting from unexpected pregnancies. The pregnant uteri were collected as part of the post-slaughter processing, with fetuses being left inside the unopened uterus. To identify the gestational age of the fetuses, the following mathematical formula has been used: GA (gestational age) = CRL × 3 + 21. Where 3 and 21 are constant values, and CRL was measured between the most rostral point of the crown and the point of the origin of the tail [29,30,31]. Not all harvested samples were viable for measurements and investigations, with only 8 sets of samples (fetuses) being evaluated metrically (out of 13) originating from 4 different pregnant sows, nominated as A, B, C and D (Table 1).

Following the previous research protocols, the tympanic cavities of each fetus were opened at the level of the tympanic bulla (*Bulla tympanica*), and the chain of auditory ossicles was visualized for removal. The most lateral (the malleus) was observed with its handle (*Manubrium mallei*) coming into direct contact with the tympanic membrane *(Membrane tympani*). After a careful lavage with a normal saline solution for the removal of organic debris and bony splinters, the auditory ossicles were detached. For this process, fine scalpel blades and sharp forceps were utilized. After the ligaments suspending the ossicular chain were severed, each bone was removed and preserved in 10% formaldehyde for 14 days. Because of their largely cartilaginous structure, these ossicles were notably delicate and frail, especially the most medially located stapes were especially difficult pieces to collect without damage. Images of the specimens were attained using an Olympus SZX7 dissection microscope (Olympus Surgical Technologies America, Westborough, MA, USA) with an incorporated DP27 camera. The system was associated with the cellSens^©^ 3.2 Standard Imaging Software (Olympus Surgical Technologies America, https://evidentscientific.com/en/software/cellsens accessed on 9 December 2024). These images were accentuated with fine contrast and background adjustments with the Adobe^©^ Photoshop program (San Jose, CA, USA). The morphometric analysis and measurements were completed using ImageJ 1.46 software (R.WS. ImageJ 1.46 2012) and the basic measurements feature [32].

### 2.2. Histological Technique

Using the literature-reported recommendations, the auditory ossicles of the 66-day fetuses (maternal origins A and D) were kept in 10% neutral buffered formalin for fixation for 14 days [7]. Fixated auditory ossicles were placed in labelled cassettes between biopsy pads and were immersed in RDO Rapid Decalcifier (Apex Engineering Products Corporation^©^, Aurora, IL, USA) solution for one hour. After a rigorous washing process, the ossicles were placed in a Tissue-Tek E-300 processor (Rankin Biomedical Corporation^©^, Davisburg, MI, USA, https://rankinbiomed.com/) allowing for an overnight processing period. Next, the processed samples were cut at 4 µm using a Leica RM 2155 microtome (Leica Microsystems Nussloch GmBh^©^, Nußloch, Germany) with Tissue-Tek Feather 4980 low-profile blades (Sakura Finetek USA^©^, Inc., St. Torrance, CA, USA, https://www.sakuraus.com/). Finally, the Harris hematoxylin and eosin protocol (H&E) [33] was then used to stain the samples. To examine the slides, an Olympus light microscope (Olympus Corporation of Americas, Breinigsville, PA, USA) was used. The imaging acquisition process was performed with an Olympus DP 26 digital camera (Olympus Surgical Technologies America) and incorporated the cellSens 3.2 Standard image analysis software.

### 2.3. Morphometric Aspects and Measurements

Measurements of the ossicles were attained following a standardized and widely accepted systematic approach, which previous researchers have well described in otology studies of humans and other primates [34] (Table 2, Table 3 and Table 4).

Special emphasis is placed on some measurements that are considered functional lengths [35,36] (especially in the case of the malleus and incus) that are often used in calculating lever ratios and descriptions of the mechanical advantage associated with the lever system in the ear.

For area calculations, the elliptical tool function within the ImageJ^©^ 1.46 software was used to overlap the tool onto a digital image and thus automatically calculate the surface of the measured surface.

The standardized measurements and the measurement protocols have been previously described for each of the ossicles [35]. All measurements were taken by one author using the same method of measurement and calibration using ImageJ 1.46 software on digital images. These measurements resulted in a type A uncertainty data repeatability test. All photographs were taken by one author (to reduce bias). Measurements were taken from seriate images and different angles. This approach made some measurements impossible due to the different perspectives of the anatomical piece. Data were recorded by hand and then imported into a Google calculation sheet (allowing the direct calculation of indices). Data were recorded separately using the identification of an individual. Basic statistical processing was carried out with Google Sheets’ simple statistics feature, the XL Miner Analysis Toolpack add-on 2023 (FrontlineSolvers, Incline Village, NV, USA—www.solver.com (accessed on 11 December 2024) and the XLSTAT 2024.3 Clud app. For the seriate measurements, we applied the Shapiro–Wilk test (https://www.statskingdom.com/shapiro-wilk-test-calculator.html accessed on 11 December 2024) to attempt to validate measurements for normality (significance level (α) set at 0.05).

## 3. Results

The largest and most lateral of the auditory ossicles is the malleus (*malleus*) (Figure 1 and Figure 2). Its particular morphological features allow us to observe and report the same segments as in an adult specimen: the head of the malleus (*Caput mallei*), the neck of the malleus (*Collum mallei*) and the last segment, the handle of the malleus (*Manubrium mallei*) (Figure 1 and Figure 2).

The head of the malleus (*Caput mallei*) is ovoid and located medially. On the caudal surface of the head, the articular surface (*Facies articularis*) of the malleus has a saddle-shaped contour and is covered with cartilage for articulation with the corresponding articular surface on the body of the incus (Figure 3).

Laterally, the neck of the malleus (*Collum mallei*) can be described as possessing a relatively long cylindrical shape. (Table 5, Table 6, Table 7 and Table 8). Three bony processes were noted on this portion of the bone: the rostral process (*Processus rostralis*), the muscular process (*Processus muscularis*) and the lateral process (*Processus lateralis*) (Figure 1 and Figure 2).

The anterior process (*Processus rostralis*) in these subjects had a triangular shape located between the malleus and the tympanic ring on the ventrolateral aspect of the neck of the malleus. The presence of this process was confirmed in all the specimens studied. The space between the ventromedial margin of the malleus head and the medial edge of the rostral process was completed by a thin triangular bony lamina (Figure 1 and Figure 2).

The muscular process (*Processus muscularis*) of the malleus was visually obvious in development and with a conical shape. Topographically, this process was located on the caudo-medial surface of the ventral border of the neck of the malleus, near the proximal end of the handle of the malleus, and had an oblique direction relative to the manubrium. This process serves as an insertion point for the tensor tympani muscle tendon (*M. tensor tympani*), which could be partially visualized.

The lateral process of the malleus *(Processus lateralis*) (Figure 1 and Figure 2) creates a landmark between the neck and the handle of the malleus. With a triangular contour, the lateral process of the malleus comes in direct contact with the tympanic membrane.

The next segment of the malleus, following in a ventral and cranial direction, is the handle of the malleus, which makes direct contact with the tympanic membrane. From the neck of the malleus to the handle of the malleus was marked by a nearly right angle (average value 82 degrees) (Table 5).

In a transverse section, we observed that the proximal end of the manubrium, near the area of the lateral process, had a roughly triangularly shaped cross-section, while its distal extremity became more flattened. The distal end of the manubrium mallei was slightly curved in all investigated specimens (Figure 1 and Figure 2).

The histological structure of the malleus, observed in the 66-day pig fetuses, reveals two distinct cortical fascicles of mineralized hyaline cartilage (internal and external) that start in the head area and run distally towards the handle junction. At the level of the malleal head and neck, the internal cortical fascicle is larger than the external cortical fascicle. Rostrally, these two fascicles do not cover the articular surface of the malleal head where articular cartilage was documented instead. Following the head and neck areas, the endosteal surfaces of the mineralized cortical fascicles surround a large medullar cavity. This cavity allowed the identification of endochondral unmineralized cartilaginous trabeculae. Both cortical fascicles were highly vascularized in the area of the neck (Figure 3).

In the handle region, both fascicles progress downwards and touch the distal end of the manubrium. A significant difference can be observed between the internal and external fascicles at this level, the external being bigger compared to the internal one. Multiple marrow cavities with compartmentalization were observed in this segment. Note that the inner trabeculae show dense mineralization compared with similar structures described in the neck area. Starting from the lateral process and continuing distally, the perichondral surface of the external cortical fascicle is covered by the tympanic membrane.

The incus (*Incus*) (Figure 4) was the second auditory bone scrutinized. It is bordered laterally by the malleus (Figure 1 and Figure 2) and medially by the stapes in life. Anatomo-topographically, this acoustical bone was found at the level of the epitympanic recess of the middle ear. Here, we were able to identify the incudo-malleal joint, which was already formed in these pig fetuses.

The ossicle has a rectangular body (*Corpus incudis*) and two well-developed processes: the long process (*Crus longum*) and the short process (*Crus breve*) (Figure 4). In an opposite direction concerning the long and short incudal processes, the body of the incus possessed a saddle-shaped articular surface for the malleo-incudal joint.

Macroscopically and morphometrically, the two crura were evidently quite different in presentation. The short process (*Crus breve*) had a conical aspect continuing the dorsal border of the incus, standing fixed into a bony fossa of the posterior wall of the tympanic cavity. The second process, (*Crus longum*), was thinner and its total length was longer (Table 6). To ensure that the identification between these two processes was correct, an examination of the distal segment of the long process was necessary. In this region, a curved aspect was described, and its distal end bears an oval bony process for articulation with the stapes head. This process is the lenticular process (*Processus lenticularis*) (Figure 4), identified in all specimens of this study.

Histological analysis of 66-day-old pig fetuses revealed that the distal extremity of the long process of the incus and the adjacent body region beneath the articular surface exhibited comparable levels of mineralization, with the presence of two cortical fascicles: the dorsal fascicle linking the dorsal margin of the short process to the dorsal body wall as well as the long process of the incus, and the ventral fascicle connecting the ventral body wall to the long process. Both fascicles are thicker at the distal end and near the lenticular process, tapering off towards the body area and the short process (Figure 5).

A compartmentalized medullary cavity is present between the identified fascicles. They feature an internal chamber that reveals various cartilaginous trabeculae with different levels of mineralization, with the highest mineralization found near the distal end of the incus’s long process, while the body and short process areas display only small islands of mineralization.

In a lateral-medial order, the stapes (*Stapes*) (Figure 6) is the third ossicle of the auditory chain and also the smallest. Its proximal end participates in the incudostapedial joint, where the stapes articulates with the lenticular process of the incus. Its distal extremity covers the oval window (*Fenestra vestibuli*) of the petrous part of the temporal bone.

In investigated pig fetuses, the stapes had a rectangular shape, and the following segments were identified: the head of the stapes (*Caput stapedis*), the anterior crus of the stapes (*Crus rostrale*), the posterior crus of the stapes (*Crus caudale*) and the footplate of the stapes or the base of the stapes (*Basis stapedis*).

The most lateral segment was identified as the head of the stapes, and it was associated with a convex articular surface. On the caudal surface of the head (or the proximal end of the posterior crus), we identified a smooth surface which serves as the insertion point for the tendon of the stapedial muscle (*M. stapedius*). This muscle was well developed in all our subjects.

In the medial direction, the head of the stapes was continued by the anterior and posterior crura of the stapes, which connect the head with the base of the stapes. Because of the rectangular aspect of the stapes, the two crura were almost parallel, and the lengths of these structures were almost the same (Table 9 and Table 10).

The space between the anterior and posterior crura, the intercrusal foramen, had an oval shape and was covered by a thin membrane. The diameter of the intercrusal foramen was found to be in direct relation to the fetus’s age and development. It was very small and almost closed in the youngest fetuses and well-developed in the older fetuses. The distal end of the stapes, the footplate, covers the fenestra vestibuli.

At the level of this junction between the oval window and the footplate, we could identify the presence of the annular ligament (*Lig. Anulare stapedis*) which was well-developed and made the extraction of the stapes challenging. Overall, the footplate has an ellipsoidal shape.

Histological investigation of the stapes (Figure 7) allowed us to observe that the first areas of mineralization developmentally were the anterior and posterior crura, but this phenomenon stops proximally, before reaching the head area, and distally above the stapes base area. Furthermore, a lack of medullary cavities was observed in both crura. In 66-day-old pig fetuses, the head of the stapes had a typical cartilaginous structure, and the process of mineralization had not started yet. A perichondral layer was identified, and proximally to the head of the stapes, it continued as the articular capsule of the incudostapedial joint.

The footplate area revealed a compact mineralized fascicle located on its vestibular side and an unmineralized fascicle on its auditory side. Between them, cartilaginous endosteal trabeculae were observed. The footplate also possessed multiple marrow cavities (Figure 7).

## 4. Discussion

### 4.1. Morphology and Morphometry of the Fetal Auditory Ossicle

Although this study used fetal ossicular tissues of different age development (Table 1), the presence of the ossicular chain could be observed in all of the investigated specimens, similar to the reported data in human fetuses [10,11,12,13,37,38,39,40], bovine fetuses [2] and sheep fetuses [1,30].

The anatomical and topographical aspects of the ossicular chain in pig fetuses were similar to the reported data in other mammalian species, with the malleus in the most lateral position, the incus in an intermediate position and the stapes in the most medial position [7].

Special features related to the number of acoustical bones have been reported in rodents, such as chinchillas, pacas, degus, and guinea pigs. In these species, the first two auditory ossicles are fused and form an individual bone called the malleo-incal complex [23,41]. The presence of this fusion in mice or humans has a direct correlation with loss of hearing in certain conditions [9,18,19]. The presence of a supplementary bone, called os lenticulare, was reported in young specimens [2,27,30,42,43,44]. In bats and insectivores [43], the lenticular process has been described as a small cartilaginous disk attached to the long process of the incus. Even though the presence of the lenticular bone is related to the early development of the individual in this study, for all the specimens, the lenticular process appeared as a slightly enlarged structure attached to the distal end of the long process of the incus. It was not an individual bone for any of the subjects. Our findings correlate with previous data reported in fetal pigs [27] and sheep [1,30].

The oval shape of the head of the malleus reported here was also noted in previous studies in sheep fetuses [1] and bovine fetuses [2]. It seems that this oval shape of the malleus’ head is a common shape for many species [1,2,18,19,28,44,45]. A spherical/round shape of the malleus head was reported in moles [1,19,45] and a bullet-shaped/ellipsoidal-shaped malleal head was reported for the subterranean caviomorph species [46].

In our investigated specimens, the relatively long neck of the malleus (Table 5, Table 6 and Table 11) was associated with three bony processes: the anterior, the muscular and the lateral processes. A long neck of the malleus was previously reported in New Zealand rabbits [47], while a short neck was reported for mice, hamsters, sheep fetuses, European badgers [19] and African green monkeys [7]. Similar findings have been reported in human fetuses and newborns [1,38], as well as in platyrrhines [48]; however, this particular segment was not identifiable or difficult to pinpoint in water buffaloes and oxen [49].

The existence of all three bony processes associated with the neck of the malleus in our specimens has been reported in adult pigs [28], miniature pigs [37] and a large number of other species [44]. This differentiates the studied species from those reported in most primates [48,50] and the small Indian mongoose [43], wherein the muscular process is located on the inner side of the manubrium.

The morphological aspects of the rostral process of the malleus, reported by previous researchers, seem variable in nature. Here, we report a triangular shape in pig fetuses; however, it was reported as a “rose-thorn-shaped” prominence in ruminants and as an unclear process in hamsters and completely replaced by a bony lamina in sheep fetuses [1,30].

A well-developed anterior process was described in the white rhinoceros, such that the authors reported a Y-shape for the malleus in that species [4]. In human embryos, the appearance of the anterior process has been reported in 30 mm embryos (8 weeks old) and is completely ossified starting with the 10th week of gestation but is not yet attached to the malleus at that time. This process reportedly happens after week 19 [13,38]. The anterior process showed a large variability in human newborns [38]. Current data seem to suggest that during the fetal period, the anterior process is very long but decreases in length after birth [1,38,51].

The muscular process of all specimens in this study was well-developed and conical in appearance. The muscular process [28] serves as an insertion point for the distal tendon of the tensor tympani muscle [28]. In sheep fetuses, the height of this process has a strong correlation with gestational age [1,30]. The conical shape of the processus muscularis reported here is similar to the features reported in ruminants, carnivores and pigs [18,19,28,44]. In foxes, the muscular process runs in a ventral direction from its origin and parallel with the malleal manubrium [52]. This process seems to be lacking entirely in the donkey [44].

The lateral process can be used as a landmark between the *pars flaccida* and *pars tensa* of the tympanic membrane in humans and adult pigs [28]. The triangular shape of the lateral process, reported here in pig fetuses, seems to be similar to the macroscopic findings reported in sheep fetuses [1]. In foxes [52] and wolves [53], however, the other two processes of the malleus were longer [10,38]. The lateral process in human newborns and fetuses has a reportedly huge variability [10,38] and ends its development around week 34 [13] (Table 11).

The distal segment of the manubrium mallei revealed a slight curvature in all the specimens studied. Similar descriptions have been reported for human newborns [38,39] and the white rhinoceros [4]. Previous studies on human and sheep fetuses [1,10] reported the absence of distal curvatures in this age category [1,10].

Although the cross-sectional shape of the manubrium mallei in pig fetuses had a triangular shape, a few large ruminants [49] have been reported to manifest a rectangular shape in this area [49].

Structural features of the malleus described in 66-day-old fetuses are similar to the reported data [13,54] of 21–23-week-old human fetuses [13,54]. The highest proportion of mineralization was observed at the distal end of the manubrium, which makes these authors believe that the process of ossification starts at this level and progresses proximally. This theory is in accordance with reported data from 24-week-old human fetuses [54,55].

The presence of the marrow cavities reported in our specimens was similar to reported data in humans [56] and African green monkeys [7]. In a previous study of normal infant auditory ossicles [39], the presence of the marrow cavity and inner trabeculae within the malleus and incus was confirmed using micro-CT [39]. Furthermore, the presence of the Haversian canals in the malleal head has been documented in adult human specimens [54,56]. Although the histological samples allowed us to identify blood vessels in the head and neck areas of our specimens, micro-CT images did not confirm the presence of any nutrient foramen on the surface of auditory ossicles from normal human infants [39].

This research corroborates the shape of the incus (Table 12) as similar to that of previous research describing the incus as a biarticulated molar tooth [20,47,52]. The long crus of the incus (Table 7, Table 8 and Table 12) is longer than the short crus and bears the lenticular process. Similar information has been reported in roe deer, dromedary camels, rabbits, rats and chinchillas [18,49]. In goats, water buffaloes, gazelles, adult sheep and miniature pigs, the lengths of the long and short processes of the incus were also quite similarly characterized [1,37,44,52]. In human fetuses, the long process of the incus is the earliest point of ossification of the auditory ossicles, beginning at the 14th week of pregnancy [13].

The existence of mineralized cortical fascicules in 66-day pig fetuses can be compared to that which has been reported in human fetuses beginning at week 16 of gestation. To place this in perspective, a similar level of development in human fetuses is typical for 21-week-old fetuses [13]. The histological differences in the level of mineralization of the long and short crura reported here seem to be equally obvious in adult humans, as previous authors [54,56] have reported that the short crus contained a proportionate number of cartilaginous trabeculae and a reduced amount of compact bone compared to the long crus [54,56].

The morphological, morphometrical (Table 13) and topographical features of the stapes of this study were similar to the classical reported data [3,42]. The rectangular shape of the stapes measured in the pig fetuses confirmed the reported features in adult pigs [28] and was similar to the reported aspects in horses, adult ruminants and sheep fetuses [1,2,44]. Other reports have described a stirrup shape for bovine fetuses, some rodents [2] and red foxes [52] or a triangle to trapezoid shape in horses and hamsters [1,30]. In human newborns [38] and human fetuses [57], the stapes showed the largest variability of all the auditory ossicles.

The specimens studied allow for identifying the insertion point of the stapedial muscle on the caudal surface of the proximal extremity of the stapes. Our findings confirm the previously reported data in bovine fetuses [2], sheep fetuses [1], white rhinoceros [4] and humans [58], where the muscular process was described as a smooth surface [18,59]. It seems that the well-developed nature of this tuberosity in adult ruminants [44] can be correlated with the permanent activity of the stapedial muscle, which starts after birth [2].

The two crura of the stapes seem to have reliably different features depending on the species studied [20]. The rectangular aspect of the stapes reported in pig fetuses can be related to the length of the stapedial crura (Table 12 and Table 13). As previously described, both crura showed similar lengths in pig fetuses. Similar data have been reported in bovine fetuses and adult pigs, donkeys, camels and red deer [1,2,18,28,59,60]. A longer posterior crus has been reported in goats, chinchillas, humans and hamsters [1,2,20,59] and the opposite has been reported in sheep fetuses and rabbits [1].

The oval space between the two crura of the stapes showed a huge variability, seemingly related to the gestational age of specimens in this study. Similar data have been reported [57], which described different shapes of the *foramen intercusale* in human fetuses and varying by gestational age [57]. The shape of this bone has a reported wide range of variability from species to species and possesses major surgical importance in humans [19,58].

Structural aspects of the stapes described here in 66-day pig fetuses are similar to the reported data for human fetuses of 24 weeks. Similar to humans, fetal pigs at 66 days also have mineralized crura. Interestingly, the marrow cavity described previously [13,54] in 24-week human fetuses [13,54] was not detected in the specimens of this research. Previous studies have reported a parallel between the morphogenesis of the stapes and the morphogenesis of the tibia [11].

### 4.2. Adult vs. Fetal Dimensional Features of the Auditory Ossicles

The literature points out the fact that in humans, the ossicles tend to modify in peri- and post-natal development stages, with the malleus being noted as the most prone to different changes (especially concerning the manubrium that possibly curves and shortens) [38,57] as the incus is the element that changes the least post-birth.

Previous studies indicate that measurements of the malleus are the most relevant from a metrical perspective; however, those of the incus and stapes are less reliable, especially when targeting sex or side differentiation [34,35]. The dimensional variation of fetal ossicles has been described previously. An interesting approach attempts to point out the degree of development of the ossicular chain during embryonic life and the amount of development at the moment of birth. Most of the studies focus on humans [38,39,40,57] but few concentrate on the fetal features of this ossicular chain [1,2,4,38,61].

Our newly presented data (Table 10, Table 12 and Table 14) add new knowledge and metric data for the species investigated. Available comparative data for swine were available from two sources, investigating and referencing miniature pig adults [37] and adult pigs [27,28,62]. Despite the different means of measurement and different nominations, we extracted the data available for comparison in this investigation after an extensive literature search. Relative to the malleus, the only previous measurement used was the length of the manubrium; for the incus, the length of the short process has been described (that has its equivalent in the width of the malleus) and the functional length of the long process (that is listed as the “effective lever”). As for the stapes, the only reference available provided the full height and the width of the bone (Table 9, Table 10 and Table 13).

For the length of the manubrium (Table 10 and Table 14), based on the few comparative elements, a notable 80–84% dimension is observed for the fetal manubrium compared to the length of the adult. For the incus, a much higher difference is to be noted. The fetal elements seem to have reached only 55–60% of their adult size. Surprisingly, the dimensions of the stapes reach 88% and 93%, respectively, of the adult specimens comparatively.

### 4.3. Middle Ear Lever Ratio

Generally, the lever ratio compares the functional lengths of the malleus (Table 5, Table 6 and Table 15) and incus (Table 7, Table 8 and Table 15) based on their rotational axis inside the tympanic cavity. In this way, the authors proposed a standard set of measurements that will characterize the necessary details of the malleus and incus, based on the estimation of the rotational axis of the malleo-incal joint. This assembly plays a key role not only in the transmission of vibrations but also in providing mechanical advantages, thus being a relevant physiological variable in the modeling of audition [35,63,64]. The functional length of the malleus (considered to be the manubrium length-measurement (2) (Table 1, Table 5 and Table 6) and the length of the incus (considering the functional length of the long process measurement 11) (Table 2, Table 7 and Table 8) are the reference values for the calculation of the lever ratio [35] (Table 15). As noted, the average value for the pig fetuses is 2.64, and adult individuals have a value of 1.6–1.7.

The limitations given by the early age of the investigated individuals, the limited functionality of the ossicular complex [65,66,67] and the proven dimensional difference from adult-sized specimens (Table 6 and Table 7) make this lever ratio calculation an indicator of the growth process until the moment of birth (and maybe continuing), as suggested by the comparative data from adult individuals (Figure 8).

## 5. Conclusions

Similar to most domestic mammals, the ossicular assembly in fetal pigs consists of a standard set of three ossicles: the malleus, the incus and the stapes. Macroscopically, we concluded that all structural components of the auditory chain have been well developed and allowed the identification of all the anatomical structures reported in the adult specimens. The metrical elements of the ossicles are described, and basic indices are calculated.

The malleus displays the typical morphology: A head with an ovoid aspect and a medial placement, supported by a cylindrical neck with three processes. This segment is linked with the handle of the malleus at almost a right angle. The histological structure of 66-day embryos highlighted the two distinct cortical fascicles of mineralized hyaline cartilage with different degrees of representation at the distinct parts of the bone, displaying marrow cavities with compartmentalized areas, and highly mineralized at the level of the manubrium.

The incus, part of the incudo-malleal joint, already formed in these pig fetuses, displays a rectangular body with a saddle-shaped articular surface and two processes: the long process and the short process, slightly different in dimension. Structurally, the two cortical fascicles were visible. The body and the long process displayed a significant level of mineralization in 66-day fetuses.

The stapes have a typical rectangular shape, with a head, two almost even and parallel crura and a base. The area of the stapedial muscle insertion is visible on the caudal surface of the head. The ellipsoidal footplate of the stapes shows the existence of the annular ligament. The histological investigation highlighted mineralized areas present at the level of the crura in the 66-day embryos with no medullary cavities. The footplate shows a compact mineralized fascicle on its vestibular side as the auditory surface is unmineralized, separated by the latter by a cartilaginous endosteal trabecular layer.

Our findings indicate that, while considering the development of the auditory ossicles, the malleus and stapes achieve approximately 80% of their developmental stage at 60% gestational age, in contrast to the incus, which has not yet reached 50–60% of its adult size by the end of gestational stage III in swine. Comparatively, human ossicular morphology suggests a comparable length development of roughly 60–65% at 20–24 weeks of gestation [13].

The current findings can serve as a starting point for future research projects focused on the embryological development of the auditory ossicles in different domestic mammalian species.

## Figures and Tables

**Figure 1 animals-15-01129-f001:**
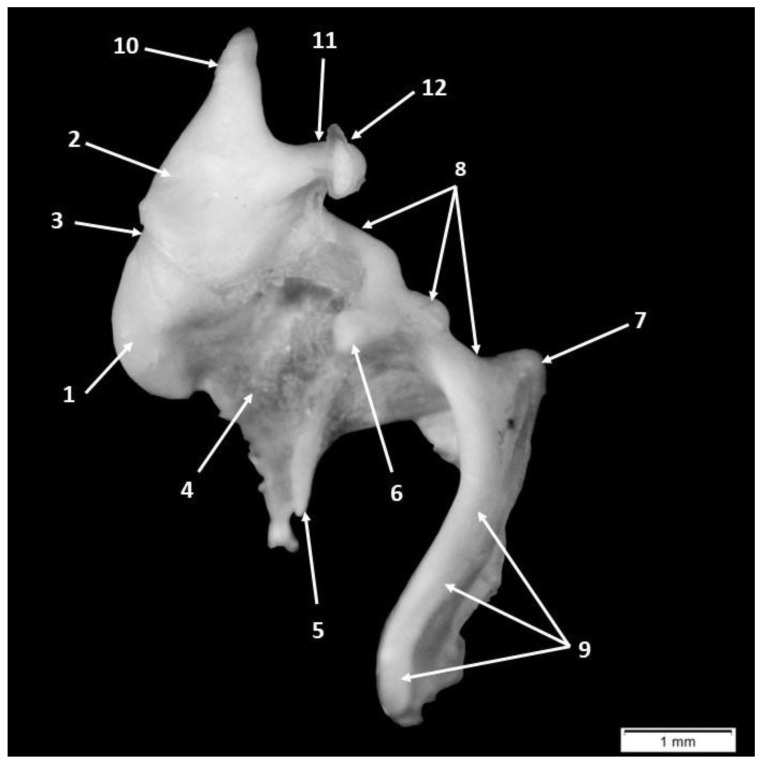
Incus and malleus joined in the incudo-malleal joint: 1. head of the malleus, 2. body of the incus, 3. incudo-malleal joint, 4. bony lame, 5. anterior process of the malleus, 6. muscular process of the malleus, 7. lateral process of the malleus, 8. neck of the malleus, 9. handle of the malleus, 10. the short process of the incus, 11. the long process of the incus and 12. lenticular process.

**Figure 2 animals-15-01129-f002:**
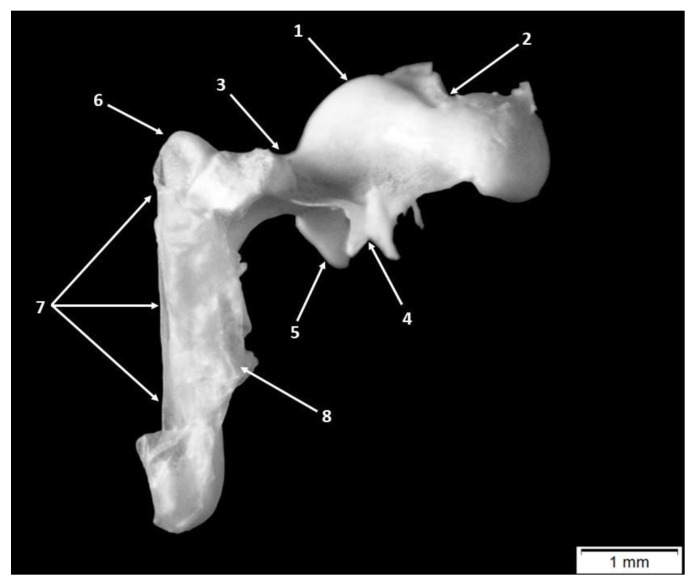
Malleus—anatomic features. 1. head of the malleus, 2. articular surface, 3. neck of the malleus, 4. anterior process, 5. muscular process, 6. lateral process, 7. handle of the malleus and 8. tympanic membrane.

**Figure 3 animals-15-01129-f003:**
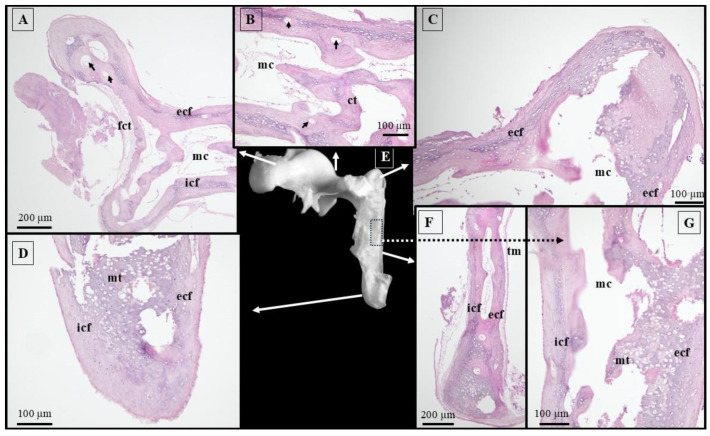
Histological features of the malleus in 66-day-old pig fetuses. (**A**) malleal head area/H&E staining; (**B**) malleal neck area/H&E staining; (**C**) lateral tubercle/H&E staining; (**D**) malleal handle bottom extremity/H&E staining; (**E**) malleus, macroscopical features; (**F**) malleal handle/H&E staining; (**G**) malleal handle, proximal end/H&E staining; ct—cartilaginous trabeculae; ecf—external cortical fascicles; fct—fibrous connective tissue; icf—internal cortical fascicles; mc—marrow cavity; mt—mineralized trabeculae; tm—tympanic membrane; and black arrows—blood vessels.

**Figure 4 animals-15-01129-f004:**
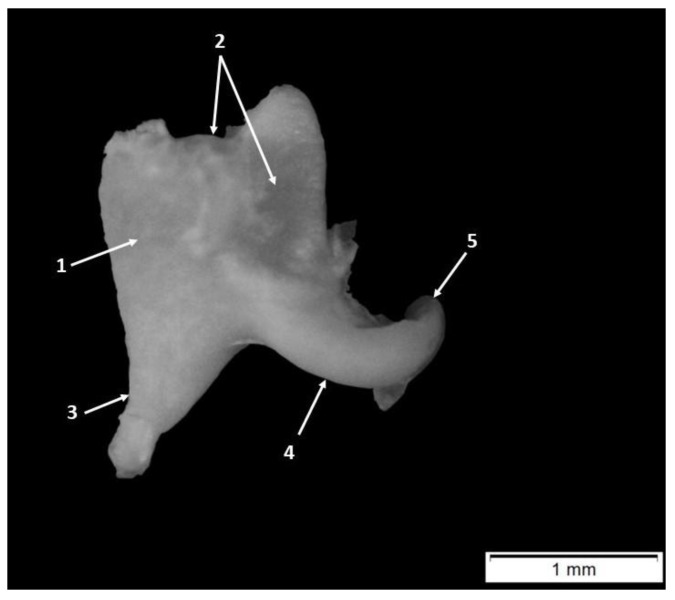
Incus—anatomic features. 1. Body of the incus, 2. articular surface of the incus, 3. short process of the incus, 4. long process of the incus, and 5. lenticular process.

**Figure 5 animals-15-01129-f005:**
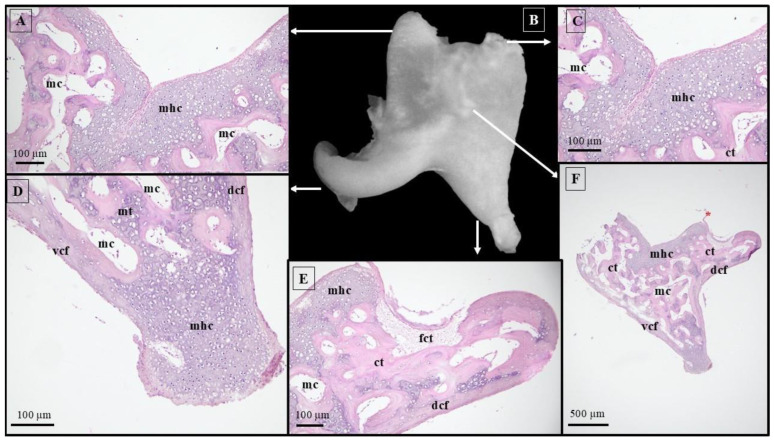
Histological features of the incus in 66-day-old pig fetuses. (**A**) Incudal articular surface/H&E staining; (**B**) incus, macroscopical features; (**C**) incudal articular surface/H&E staining; (**D**) long process/H&E staining; (**E**) short process/H&E staining; (**F**) incudal body/H&E staining; ct—cartilaginous trabeculae; dcf—dorsal cortical fascicles; fct—fibrous conjunctive tissue; mc—marrow cavity; mhc—mineralized hyaline cartilage; mt—mineralized trabeculae; vcf—ventral cortical fascicles; and red asterisk—malleo-incal joint capsule.

**Figure 6 animals-15-01129-f006:**
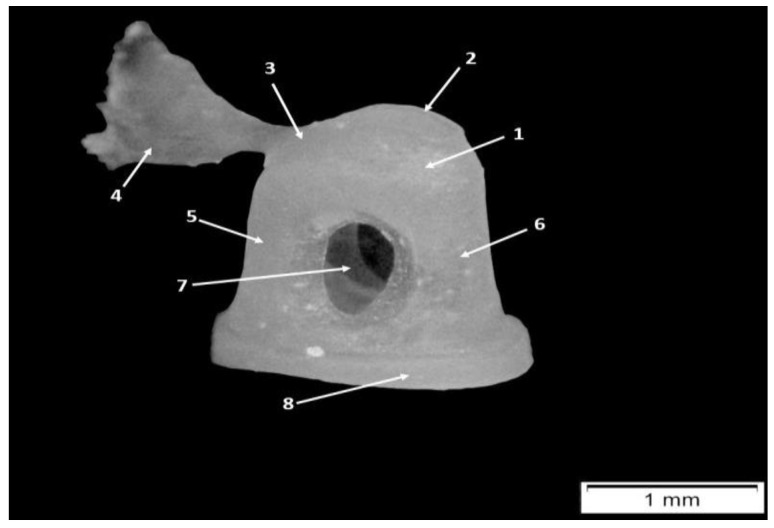
Stapes—anatomical features. 1. Head of the stapes, 2. articular surface, 3. insertion surface for the stapedial muscle, 4. stapedial muscle, 5. posterior crus of the stapes, 6. anterior crus of the stapes, 7. intercusal foramen, and 8. stapedial footplate of the stapes.

**Figure 7 animals-15-01129-f007:**
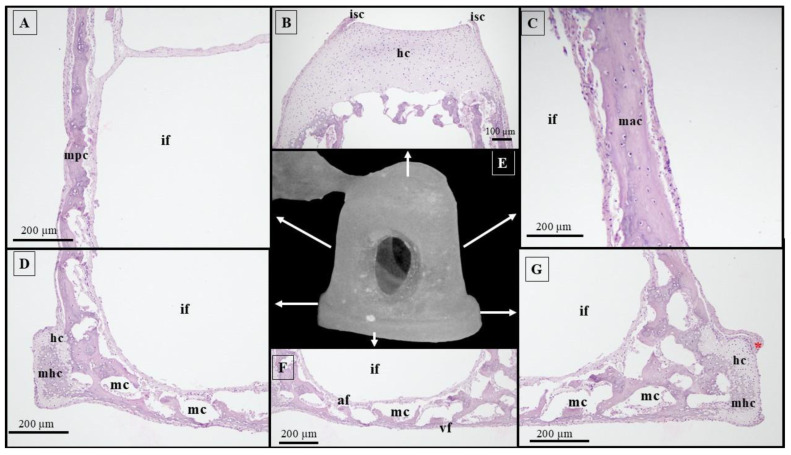
Histological features of the stapes in 66-day-old pig fetuses. (**A**) Posterior crus/H&E staining; (**B**) head of the stapes/H&E staining; (**C**) anterior crus/H&E staining; (**D**) caudal segment of the footplate/H&E staining; (**E**) stapes, macroscopical features; (**F**) footplate/H&E staining; (**G**) rostral segment of the footplate/H&E staining; mpc—mineralized posterior crus; mac—mineralized anterior crus; af—acoustical fascicle of the footplate; if—intercrural foramen; isc—incudostapedial joint capsule; hc—hyaline cartilage; mc—marrow cavity; mhc—mineralized hyaline cartilage; vf—vestibular compact fascicle of the footplate; and red asterisk—annular ligament.

**Figure 8 animals-15-01129-f008:**
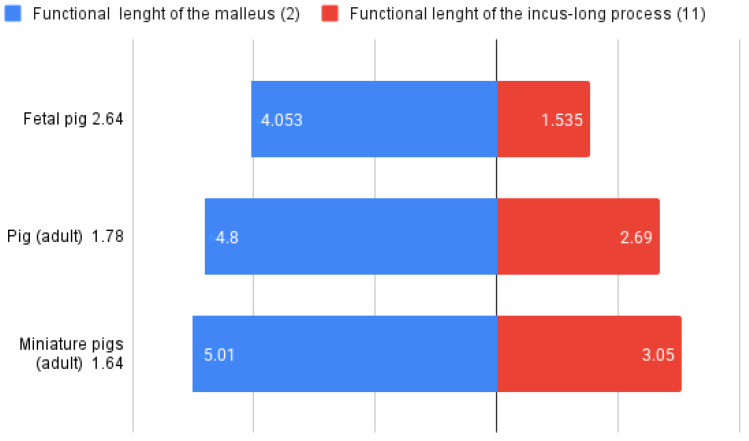
Graphical representation of the functional length of the malleus and functional length of the incus for the reference species.

**Table 1 animals-15-01129-t001:** Evaluated gestational age and origin for the studied pig fetuses used in this study.

Embryo Number	Estimated Gestational Age	Maternal Origin
1	48 (days)	B
2	66 (days)	D
5	72 (days)	C
6	66 (days)	D
8	66 (days)	A
9	66 (days)	A
10	66 (days)	A
13	66 (days)	D

**Table 2 animals-15-01129-t002:** Measurement protocol and indices for the malleus [35].

*X* axis	Midpoint of the minimum neck width—the most noticeable point along the top of the head
*Y* axis	Most inferior point of the short process and the manubrium:
1 Total length	tip of the manubrium to the top of the head
2 Manubrium length	Tip of the short process to the tip of the manubrium following *X* axis
3 Manubrium M-L thickness	M-L thickness of the manubrium at mid-manubrium length, perpendicular to *X* axis
4 Manubrium arc depth	Maximum depth of the curvature of the arc of the manubrium, following *X* axis
5 Corpus length	Tip of the head to the lower border of the manubrium following *X* axis
6 Neck width	Anterior and posterior borders of the neck
7 S-L head width	Maximum distance between 2 parallel lines marking the widest points of the margin of the head, taken following the *X* axis
8 Angle between axes	X-Y angle
Manubrium/length index	(manubrium length/total length) × 100
Manubrium robusticity index	(manubrium ML thickness/corpus length) × 100
Manubrium/corpus index	(manubrium length/corpus length) × 100
Corpus/length index	(corpus length/total length) × 100

**Table 3 animals-15-01129-t003:** Measurement protocol and indices for the incus [35].

*X* axis	Line that joins the most salient point along the anterior portion of the superior border of the body
*Y* axis	Line that joins the tip of the long process to the most salient point along the superior border of the body
*Z* axis	Line joining the tip of the long process to the most external point along the margin of the anterior facet
9 Short process length	Maximum distance from the tip of the short process to the most salient point along the anterior portion of the superior border of the body, following *X* axis
10 Long process length	Maximum distance from the tip of the long process to the most salient point along the superior border of the body
11 Functional length of the long process	Perpendicular distance from the *Z* axis (rotational axis) to the tip of the long process
12 Arc depth of the long process	Maximum depth of the arc along the long process measured from the plane defined by the lateral outmost point along the tip of the long process
13 Articular facet height	Max height of the articular facet with the bone oriented along the rotational axis
14 Angle between the axes	Angle formed by the *X* and *Y* axes
15 Interprocess length	Maximum distance between the most salient points along the superior margin of the short process and the tip of the long process
16 Interprocess arc depth	Maximum depth of the curvature between the short and long process tips
Incudal index	9/10 × 100
Long process index	11/10 × 100
Relative articular facet height	13/10 × 100

**Table 4 animals-15-01129-t004:** Measurement protocol and indices for the stapes [35].

*X* axis	Line joining the antero-superior corner of the footplate and the tip of the head
*Y* axis	Line joining the posterior-superior corner of the footplate and the tip of the head
*Z* axis	Line joining the most inferior points along the footplate margin anteriorly and posteriorly
19 Total height of the stapes	Maximum height from the lower margin of the footplate to the tip of the head perpendicular to the *Z* axis
20 Head height	Minimum distance between the superior margin of the obturator foramen and the top of the head, taken perpendicular to the *Z* axis
21 Obturator foramen height	Maximum height of the obturator foramen taken perpendicular to the *Z* axis
22 Obturator foramen width	Maximum width of the obturator foramen taken parallel to the *Z* axis
23 Maximum width of the crura	Maximum width across the anterior and posterior crurae, taken on the external aspect and parallel to the *Z* axis
24 Posterior crus length	Maximum distance from the posterior-superior corner of the footplate to the tip of the head, following *Y* axis
25 Posterior crus arc depth	Maximum depth of the curvature of the posterior crus taken parallel to the *Y* axis
26 Anterior crus length	Maximum distance from the anterio-superior corner of the footplate to the tip of the head following *X* axis
27 Anterior crus arc depth	Maximum depth of the curvature of the anterior crus taken parallel to the *X* axis
28 Angle A	Angle between the anterior and posterior crurae or between the *X* and *Y* axes
29 Angle B	Angle between the anterior crus and the footplate or between the *X* and *Z* axes
30 Angle C	Angle between the posterior crus and the footplate between *Y* and *Z* axes
31 Footplate length	Maximum length of the footplate
32 Footplate width	Maximum width of the footplate
33 Footplate area	Measured area of the footplate
Stapedial index	31/19 × 100
Relative head height	20/19 × 100
Obturator foramen index	21/22 × 100
Footplate index	31/32 × 100
Crural index	36/24 × 100

**Table 5 animals-15-01129-t005:** Metric dimensions of the malleus in the investigated pig individuals.

Specimen	Measurement Number	Mean Values	
1	1	5.34 mm (*n* = 4)	
	2	4.51 mm (*n* = 5)	
	3	0.58 mm (*n* = 4)	
	5	3.17 mm (*n* = 4)	
	6	0.86 mm (*n* = 4)	
	7	1.14 mm (*n* = 3)	
	8	75.52° (*n* = 3)	
2	1	5.84 mm (*n* = 3)	
	2	3.85 mm (*n* = 6)	
	3	0.44 mm (*n* = 5)	
	5	4.21 mm (*n* = 4)	
	6	1.07 mm (*n* = 2)	
	7	1,12 mm (*n* = 5)	
	8	77.01° (*n* = 3)	
5	1	5.62 mm (*n* = 13)	
	2	3.97 mm (*n* = 14)	
	3	0.45 mm (*n* = 14)	
	5	4.16 mm (*n* = 13)	
	6	0.71 mm (*n* = 13)	
	7	1.24 mm (*n* = 15)	
	8	85.2° (*n* = 13)	
6	1	6.39 mm (*n* = 2)	
	2	4.65 mm (*n* = 2)	
	3	0.48 mm (*n* = 2)	
	5	4.53 mm (*n* = 2)	
	6	0.85 mm (*n* = 2)	
	7	1.47 mm (*n* = 2)	
	8	80.25° (*n* = 2)	
8	1	5.85 mm (*n* = 2)	
	2	4.11 mm (*n* = 2)	
	3	0.40 mm (*n* = 2)	
	5	4.30 mm (*n* = 2)	
	6	0.99 mm (*n* = 2)	
	7	1.63 mm (*n* = 1)	
	8	90.4° (*n* = 1)	
9	1	-	
	2	-	
	3	3.94 (*n* = 1)	
	5	-	
	6	-	
	7	-	
	8	85.3 (*n* = 1)	
10	1	5.25 mm (*n* = 2)	
	2	3.55 mm (*n* = 2)	
	3	0.35 mm (*n* = 2)	
	5	3.98 mm (*n* = 2)	
	6	0.68 mm (*n* = 1)	
	7	1.50 mm (*n* = 2)	
	8	85.21° (*n* = 2)	

**Table 6 animals-15-01129-t006:** Summary of the dimensional variations of the malleus in the pig fetus specimens and the normality test.

Measurement Number	Mean Values	Standard Deviation	Shapiro–Wilk Test for Normality
1	56,448	347	The Shapiro–Wilk test did not show a significant departure from normality, W(27) = 0.96, *p* = 0.328
2	4.0503	0.383	The Shapiro–Wilk test did not show a significant departure from normality, W(33) = 0.95, *p* = 0.115
3	0.4633	0.103	The Shapiro–Wilk test did not show a significant departure from normality, W(30) = 0.96, *p* = 0.255
5	4.04	0.404	The Shapiro–Wilk test showed a significant departure from normality, W(27) = 0.8, *p* < 0.001
6	0.8004	0.16	The Shapiro–Wilk test showed a significant departure from normality, W(25) = 0.89, *p* = 0.013
7	1.2713	0.196	The Shapiro–Wilk test showed a significant departure from normality, W(29) = 0.9, *p* = 0.008
8	82.9857	5.532	The Shapiro–Wilk test did not show a significant departure from normality, W(26) = 0.95, *p* = 0.192

**Table 7 animals-15-01129-t007:** Metric dimensions for the incus in the investigated pig individuals.

Specimen	Measurement Number	Mean Values
2	9	1.87 mm (*n* = 2)
	10	2.305 mm (*n* = 2)
	11	1.635 mm
	12	0.145 mm
	13	1.3 mm (*n* = 1)
	14	52.4° (*n* = 2)
	15	1.87 mm
	16	0.45 mm
5	9	2.065 mm (*n* = 4)
	10	2.29 mm (*n* = 4)
	11	1.351 mm (*n* = 4)
	12	0.292 mm (*n* = 4)
	13	1.12 mm (*n* = 4)
	14	50.19° (*n* = 4)
	15	1.74 mm (*n* = 4)
	16	0.452 mm (*n* = 4)
8	9	1.98 mm (*n* = 2)
	10	2.37 mm (*n* = 2)
	11	1.62 mm (*n* = 2)
	12	0.1 mm (*n* = 1)
	13	1.04 mm (*n* = 2)
	14	52.6° (*n* = 1)
	15	1.96 mm (*n* = 2)
	16	0.385 mm (*n* = 2)
9	9	1.88 mm (*n* = 1)
	10	2.39 mm (*n* = 1)
	11	1.6 mm (*n* = 1)
	12	-
	13	1.23 mm (*n* = 1)
	14	53° (*n* = 1)
	15	2.21 mm (*n* = 1)
	16	0.4 mm (*n* = 1)
10	9	1,96 mm (*n* = 2)
	10	2.215 mm (*n* = 2)
	11	1.55 mm (*n* = 2)
	12	0.26 mm (*n* = 2)
	13	1.09 mm (*n* = 2)
	14	49.15° (*n* = 2)
	15	2 mm (*n* = 2)
	16	0.435 mm (*n* = 2)
13	9	1.85 mm (*n* = 3)
	10	2.49 mm (*n* = 3)
	11	1.62 mm (*n* = 3)
	12	0.2 mm (*n* = 3)
	13	1.11 mm (*n* = 3)
	14	51.72° (*n* = 3)
	15	2.17 mm (*n* = 3)
	16	0.453 mm (*n* = 3)

**Table 8 animals-15-01129-t008:** Summary of the dimensional variations for the incus in the pig specimens and the normality test.

Measurement Number	Mean Values (mm)	Standard Deviation	Shapiro–Wilk Test for Normality
9	1.95	0.111	The Shapiro–Wilk test did not show a significant departure from normality, W(14) = 0.89, *p* = 0.088
10	2.343	0.151	The Shapiro–Wilk test did not show a significant departure from normality, W(14) = 0.96, *p* = 0.654
11	1.535	0.142	The Shapiro–Wilk test did not show a significant departure from normality, W(14) = 0.91, *p* = 0.166
12	0.223	0.079	The Shapiro–Wilk test did not show a significant departure from normality, W(12) = 0.97, *p* = 0.951
13	1.123	0.108	The Shapiro–Wilk test did not show a significant departure from normality, W(13) = 0.91, *p* = 0.188
14	51.124	3.15	The Shapiro–Wilk test did not show a significant departure from normality, W(13) = 0.95, *p* = 0.574
15	1.953	0.199	The Shapiro–Wilk test did not show a significant departure from normality, W(14) = 0.92, *p* = 0.233
16	0.436	0.038	The Shapiro–Wilk test did not show a significant departure from normality, W(14) = 0.96, *p* = 0.672

**Table 9 animals-15-01129-t009:** Metric dimensions for the stapes in the investigated pig individuals.

Specimen	Measurement Number	Mean Values
6	19	2.18 mm (*n* = 1)
	20	0.78 mm (*n* = 1)
	21	0.83 mm (*n* = 1)
	22	0.86 mm (*n* = 1)
	23	1.65 mm (*n* = 1)
	24	2.09 mm (*n* = 1)
	25	0.36 mm (*n* = 1)
	26	1.98 mm (*n* = 1)
	27	0.36 mm (*n* = 1)
	28	50.7° (*n* = 1)
	29	69° (*n* = 1)
	30	61° (*n* = 1)
	31	2.02 mm (*n* = 1)
	32	-
8	19	1.92 mm (*n* = 1)
	20	0.75 mm (*n* = 1)
	21	0.87 mm (*n* = 1)
	22	0.73 mm (*n* = 1)
	23	1.65 mm (*n* = 1)
	24	2.05 mm (*n* = 1)
	25	0.4 mm (*n* = 1)
	26	1.82 mm (*n* = 1)
	27	0.31 mm (*n* = 1)
	28	60° (*n* = 1)
	29	64° (*n* = 1)
	30	52° (*n* = 1)
	31	1.78 mm (*n* = 1)
	32	-
9	19	2.08 mm (*n* = 3)
	20	0.91 mm (*n* = 3)
	21	0.89 mm (*n* = 3)
	22	0.753 mm (*n* = 3)
	23	1.57 mm (*n* = 3)
	24	2.05 mm (*n* = 3)
	25	0.42 mm (*n* = 3)
	26	1.94 mm (*n* = 3)
	27	0.396 mm (*n* = 3)
	28	53.26° (*n* = 3)
	29	67° (*n* = 3)
	30	58.33° (*n* = 3)
	31	1.83 mm (*n* = 3)
	32	-
10	19	1.89 mm (*n* = 2)
	20	0.71 mm (*n* = 2)
	21	0.745 mm (*n* = 2)
	22	0.655 mm (*n* = 2)
	23	1.58 mm (*n* = 2)
	24	1.915 mm (*n* = 2)
	25	0.465 mm (*n* = 2)
	26	1.69 mm (*n* = 2)
	27	0.48 mm (*n* = 2)
	28	58° (*n* = 2)
	29	68° (*n* = 2)
	30	55° (*n* = 2)
	31	1.685 mm (*n* = 2)
	32	-
13	19	2.07 mm (*n* = 2)
	20	0.82 mm (*n* = 2)
	21	0.88 mm (*n* = 2)
	22	0.715 mm (*n* = 2)
	23	1.59 mm (*n* = 2)
	24	1.99 mm (*n* = 2)
	25	0.415 mm (*n* = 2)
	26	1.865 mm (*n* = 2)
	27	0.47 mm (*n* = 2)
	28	53.5° (*n* = 2)
	29	66.5° (*n* = 2)
	30	57.5° (*n* = 2)
	31	1.846 mm (*n* = 3)
	32	1.08 mm (*n* = 1)

**Table 10 animals-15-01129-t010:** Summary of the dimensional variations of the stapes in the pig specimens and the normality test.

Measurement Number	Mean Values (mm)	Standard Deviation	Shapiro–Wilk Test for Normality
19	2.028	0.117	The Shapiro–Wilk test did not show a significant departure from normality, W(9) = 0.9, *p* = 0.239
20	0.815	0.089	The Shapiro–Wilk test did not show a significant departure from normality, W(9) = 0.94, *p* = 0.694
21	0.846	0.073	The Shapiro–Wilk test did not show a significant departure from normality, W(9) = 0.95, *p* = 0.806
22	0.732	0.065	The Shapiro–Wilk test did not show a significant departure from normality, W(9) = 0.95, *p* = 0.725
23	1.594	0.043	The Shapiro–Wilk test did not show a significant departure from normality, W(9) = 0.95, *p* = 0.762
24	2.012	0.105	The Shapiro–Wilk test did not show a significant departure from normality, W(9) = 0.87, *p* = 0.131
25	0.42	0.051	The Shapiro–Wilk test did not show a significant departure from normality, W(9) = 0.96, *p* = 0.910
26	1.858	0.129	The Shapiro–Wilk test did not show a significant departure from normality, W(9) = 0.97, *p* = 0.941
27	0.417	0.074	The Shapiro–Wilk test did not show a significant departure from normality, W(9) = 0.86, *p* = 0.104
28	54.836	3.198	The Shapiro–Wilk test did not show a significant departure from normality, W(9) = 0.94, *p* = 0.579
29	67	2.692	The Shapiro–Wilk test did not show a significant departure from normality, W(9) = 0.95, *p* = 0.720
30	57	2.958	The Shapiro–Wilk test did not show a significant departure from normality, W(9) = 0.94, *p* = 0.666
31	1.82	0.102	The Shapiro–Wilk test did not show a significant departure from normality, W(10) = 0.95, *p* = 0.735
32	1.08	-	Not applicable (*n* = 1)

**Table 11 animals-15-01129-t011:** Variance indices of the malleus.

Index Malleus	Value (Average/Variation)
Manubrium/length index	71.46 ± 6.342
Manubrium robusticity index	11.21 ± 4.242
Manubrium/corpus index	101.02 ± 18.551
Length index	69.35 ± 15.159

**Table 12 animals-15-01129-t012:** Variance indices of the incus.

Index	Value (Average/Variation)
Incudal index	83.621 ± 7.862
Long process index	65.727 ± 7.223
Relative articular facet height	44.312 ± 13.696

**Table 13 animals-15-01129-t013:** Variance indices of the stapes.

Index	Value (Average/Variation)
Stapedial index	89.522 ± 4.324
Relative head height	40.178 ± 3.540
Obturator foramen index	116.150 ± 11.409
Footplate index	173.148
Crural index	92.374 ± 4.077

**Table 14 animals-15-01129-t014:** The comparative metrical features for the incus, malleus and stapes.

	Pigs (Adult)	Miniature Pigs (Adult)	Fetal Pig
Malleus: length of the manubrium (mm)	4.8 ± 0.4	5.01 ± 0.5	4.05 ± 0.38
Incus: length of the short process (9), incus width	3.12 ± 0.15	2.59 ± 0.16	1.95 ± 0.111
Incus: functional length of the long process (11) or the effective lever	2.69 ± 0.15	3.05 ± 0.14	1.53 ± 0.14
Stapes: total height (19)	2.16 ± 0.14	2	2.02 ± 0.11
Stapes: footplate width (31), total width	2.05 ± 0.14	N/A	1.82 ± 0.102

**Table 15 animals-15-01129-t015:** Reference values for the calculation of the lever ratio in the studied specimens.

Specimen Number	Functional Length of Malleus (mm)	Functional Length of Long Process (mm)	Lever Ratio
1	4.51	No available data	-
2	3.85	1.635	2.35
5	3.97	1.351	2.93
6	4.65	No available data	-
8	4.11	1.62	2.53
9	No available data	1.6	-
10	3.55	1.55	-
13	No available data	1.62	-

## Data Availability

The raw data supporting the conclusions of this article will be made available by the authors upon request.

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
