# Peer review of "Morphological, Histological and Morphometrical Aspects of Auditory Ossicles in Pig Fetuses (Sus scrofa domestica)"

_animals, 2025, doi:10.3390/ani15081129_

Round 1
Reviewer 1 Report
Comments and Suggestions for Authors
It is a very important and little studied topic in the anatomy of small mammals.
The text is disorganized. Many sentences from the Materials and Methods section are in the Discussion; and many sentences from the Discussion appear in Materials and Methods. The authors should restructure the text.
Please see the comments in the attached file.

Author Response
Thank you very much for your observations. We saw the notes on the file and several corrections were made to our initial textfile.
- Correction in the title- addition of the suggested element
- The summary was reworked and structured as one paragraph with no subdivisions.
- For the keywords some of the items were changed, using Latin nominations according to NAV
- Removed the suggested paragraph (lines 60-65) and modified the final phrase, removed the citation in the final sentence
- Materials and methods: the exact age of the fetuses is not perfectly known. As you saw, these were accidental pregnancies discovered during the slaughtering process. This is why the formula of assessment of gestational age (as refernced in the text) was used (table 1). The weights of each fetus were not recorded individually. All we have is an approximate value that we prefer not to use (subjective).
- Latin terms italicized
- Supplementary explanations were offered with respect to data contained in Table 1. Minor changes in the table
- 133-135 moved to discussions
- 200-201 removed
- 238 corrected
- Discussions- lines 324-326 removed and following text flush
- Conclusions part was slightly shortened, some sentences were removed, so the basic elements of morphology described were shortened and became more concise. Left unchanged the histological description.
- 541-552 lines moved to discussion and phrasal arrangements done accordingly
- Data availability statememnt inserted
- Author contributions added/corrected
Reviewer 2 Report
Comments and Suggestions for Authors
Abstract: The first sentence in the background is vague and lacks specificity. Phrases like “was not a main topic for investigation until lately” are imprecise. While macroscopical, metrical, and histological assessments are mentioned, the specific techniques, sample size, and type of measurements are not clearly defined. Although the morphology of each ossicle is described, no numerical values or statistical data (measurements, sample sizes, variation) are provided in the abstract.p values should be written. Study group is not clear. The use of articles and prepositions also needs grammatical corrections. The conclusion lacks a clear statement of novelty or scientific implication. The abstract does not indicate why the findings are important
Introduction:The sentence "Most anatomical studies to date..." requires appropriate references to support the statement.
Results:P-values are missing both in the text and in the tables.
The statistical methods used in the analysis are not described. Please specify the analytical techniques employed.
Discussion:Tables 8–15 should be moved to the Results section, as they contain findings, not interpretations.There are major issues with the organization of the manuscript.
The Discussion should be divided into subsections, and the raw findings currently included in the discussion must be relocated to the Results section.The Discussion should focus on the interpretation of findings, their contribution to the literature, and any novel insights brought by the study.
Conclusion:This section should be shortened and presented concisely.References should not be included in the conclusion.
Outdated references (e.g., from 1971 and 1998) should be removed or replaced with more recent literature.
ImageJ cannot be cited as a reference; instead, provide a citation for the software (in the methods section, with the appropriate version).
Comments on the Quality of English LanguageThe English could be improved to more clearly express the research
Author Response
Thank you very much for your observations. We saw the notes and comments and several corrections were made to our initial textfile.
- Abstract- some changes in structure were made so they reflect better the importance of the study. Some sentences were removed- eg…until lately…and some basic numeric data about our study group was inserted. We did not add p-values as the sample is in fact quite small and the details in the maintext are illustrative as far as more basic statistics are concerned.
- Introduction- slightly changed. The suggested sentence has a bibliographic reference
- Results- as most of the basic measurements do not count more than 4-10, the statistical interpretation we considered not necessary in terms of more advanced data (see values for n). only in the Discussion part, we applied the descriptors for variation in the usage of Saphiro-Wilk test
- Discussions: as suggested, tables with the summative statistical data were moved into the Results section. A supplementary sub-heading was added in the section, as suggested, indicating the discussion on the basic morphological and morphometrical data, followed by the second dealing with comparative morphometry
- The indicated bibliographical source- Marrable 1971- we would like to keep it as is serves as a starting point for our investigation similar to Malo 1998 as they may be considered almost “classical sources” for the study of the embryology of the pig respectively the development of the middle ear
- ImageJ was listed as software in the text and another useful reference was added for the application of ImageJ
Reviewer 3 Report
Comments and Suggestions for Authors The work by Olimpiu et al. presents the results of an anatomical and histological study ofpig ear bones at various stages of fetal development. It might have been better to take
samples from a greater number of days of gestation. However, since there is little prior
evidence, the work is of interest. The introduction presents a description of the anatomical characteristics of these bones,
based especially on what is known about pigs and humans, the latter species being the basis
of the developmental data included.However, both in the introduction and later in the
discussion, aspects are missing that would make the work more interesting. In particular,
some mention would be important of the characteristics of hearing in pigs, which would
be interesting to relate to morphological aspects. Evolutionary aspects are also not
mentioned, when it is mentioned that birds and reptiles have fewer bones, which is very
important in the evolution of mammals.
The materials and methods are well described. The only thing that needs to be corrected is the tables listing the parameters analyzed (Tables 2, 3, and 4). These tables have formatting errors. The results are presented correctly with good microphotographs. The discussion has some shortcomings that require rewriting. On the one hand
(which is formal but important), in some cases the common name is written and in others
the scientific name (sometimes misspelled, for example with the generic epithet in capital
letters) of the mentioned species. I suggest writing both names the first time the
species is mentioned, but the important thing is to unify the way they are written.
On the other hand, aspects relating the characteristics of these bones to hearing
differences between species are not discussed. Nor is the ossification of the auditory
bones related to that of other bones in pigs; it is known that in humans, it occurs
particularly early in development. While the comparative anatomy study is very good, it would be necessary to provide an
evolutionary and functional context to make the work more interesting for a broader and
more diverse audience.
Author Response
Thank you very much for your observations. We saw the notes and comments and several corrections were made to our initial textfile in accordance to the other reviewer’s comments, too
In respect to yours, we would like to add some remarks:
- Our paper presents the results of the investigation carried out on 13 fetal pigs, as mentioned in the Materials and Methods section. We are totally aware that the larger the sample, the more relevant the results. This subject was, somehow, mentioned by the other reviewer. That was the available material and this was the main reason no very complex statistical investigation was carried out. The evolutionary aspects invoked by you might become interesting as several species or subspecies of the Sus genre are analyzed from this perspective. This might be the scope of a much ampler work that might appear later, as the scope widens progressively. We have in our mind the wild boar but we do not possess other samples (small or large) to approach differential aspects in terms of lever ratio or different indexes.
- Some tables were moved along the text, numbering is different and they will go through editing according to the journal’s demands. Checked the formatting again and re-applied the designated template
- Species names- revised, using mainly common names- some scientific names also italicized
- Evolutionary aspects and the ossification process remain still a goal for the present situation, as our focus was mainly on the gross and micromorphological approach, linked with the metrical collected data.
Round 2
Reviewer 2 Report
Comments and Suggestions for Authors
Necessary corrections were done
Comments on the Quality of English LanguageThe English could be improved to more clearly express the research. It needs editing.
Reviewer 3 Report
Comments and Suggestions for Authors I believe that although the discussion in an evolutionary or functional frameworkwould have significantly enriched the manuscript, and taking account that the authors realized the changes
suggested by other reviewersthe changes made improved the work and
it can be accepted.